# Provably adaptive reinforcement learning in metric spaces

**Tongyi Cao**
University of Massachusetts Amherst
tcao@cs.umass.edu

**Akshay Krishnamurthy**
Microsoft Research NYC
akshaykr@microsoft.com

## Abstract

We study reinforcement learning in continuous state and action spaces endowed with a metric. We provide a refined analysis of the algorithm of Sinclair, Banerjee, and Yu (2019) and show that its regret scales with the *zooming dimension* of the instance. This parameter, which originates in the bandit literature, captures the size of the subsets of near optimal actions and is always smaller than the covering dimension used in previous analyses. As such, our results are the first provably adaptive guarantees for reinforcement learning in metric spaces.

## 1 Introduction

In reinforcement learning (RL), an agent learns to selects actions to navigate a state space and accumulate reward. In terms of theoretical results, the majority of results address the tabular setting, where the number of states and actions are finite and comparatively small. However, tabular problems are rarely encountered in practical applications, as state and action spaces are often large and may even be continuous. To address these practically relevant settings, a growing body of work has developed algorithmic principles and guarantees for reinforcement learning in continuous spaces.

In this paper, we contribute to this line of work on reinforcement learning in continuous spaces. We consider episodic RL where the joint state-action space is endowed with a metric and we posit that the optimal $Q^\star$ function is *Lipschitz continuous* with respect to this metric. This setup has been studied in several recent works establishing worst case regret bounds that scale with the covering dimension of the metric space (Song and Sun, 2019; Sinclair et al., 2019; Touati et al., 2020). While these results are encouraging, the guarantees are overly pessimistic, and intuition from the special case of Lipschitz bandits suggests that much more adaptive guarantees are achievable. In particular, while the Lipschitz contextual bandits setting of Slivkins (2014) is a special case of this setup, no existing analysis recovers his adaptive guarantee that scales with the *zooming dimension* of the problem.

**Our contribution.** We give the first analysis for reinforcement learning in metric spaces that scales with the zooming dimension of the instance instead of the covering dimension of the metric space. The zooming dimension, originally defined by Kleinberg et al. (2019) in the context of Lipschitz bandits, measures the size of the set of near-optimal actions, and can be much smaller than the covering dimension in favorable instances. For reinforcement learning, the natural generalization is to measure near-optimality relative to the $Q^\star$ function; this recovers the definition of Kleinberg et al. (2019) and Slivkins (2014) for bandits and contextual bandits, respectively as special cases. As a consequence, our guarantees also strictly generalize theirs to the multi-step reinforcement learning setting. In addition, our guarantee addresses an open problem of Sinclair et al. (2019) by characterizing problems where refined guarantees are possible.

Our result is based on a refined analysis of the algorithm of Sinclair et al. (2019). This algorithm uses optimism to select actions and an adaptive discretization scheme to carefully refine a coarse partition of the state-action space to focus ("zoom in") on promising regions. Adaptive discretization

is essential for obtaining instance-dependent guarantees, but the bounds in Sinclair et al. (2019) do not reflect this favorable behavior.

At a technical level, the main challenge is that, unlike in bandits, we cannot upper bound the number of times a highly suboptimal arm will be selected by the optimistic strategy. Analysis for the bandit setting uses these upper bounds to prove that the adaptive discretization scheme will not zoom in on suboptimal regions, which is crucial for the instance-dependent bounds. However, in RL, the algorithm actually can zoom in on and select actions at suboptimal regions, but only when there is significant error at later time steps. Thus, in the analysis, we credit error incurred from a highly suboptimal region to the later time step, so we can proceed as if we never zoomed in on this region at all. Formally, this analysis uses the *clipped regret decomposition* of Simchowitz and Jamieson (2019) as well as a careful bookkeeping argument to obtain the instance-dependent bound.

## 2   Preliminaries

We consider a finite-horizon episodic reinforcement learning setting in which an agent interacts with an MDP, defined by a tuple $(\mathcal{S}, \mathcal{A}, H, \mathbb{P}, r)$. Here $\mathcal{S}$ the state space, $\mathcal{A}$ is the action space, $H \in \mathbb{N}$ is the horizon, $\mathbb{P}$ is the transition operator and $r$ is the reward function. Formally, $\mathbb{P} : \mathcal{S} \times \mathcal{A} \to \Delta(\mathcal{S})$ and $r : \mathcal{S} \times \mathcal{A} \to [0, 1]$ where $\Delta(\cdot)$ denotes the set of distributions over its argument.[1]

A (nonstationary) policy $\pi$ is a mapping from states to distributions over actions for each time. Every policy has non-stationary value and action-value functions, defined as

$$V_h^\pi(x) := \mathbb{E}_\pi \left[ \sum_{h'=h}^{H} r_{h'}(x_{h'}, a_{h'}) \mid x_h = x \right], \qquad Q_h^\pi(x, a) := r_h(x, a) + \mathbb{E}\left[ V_{h+1}^\pi(x') \mid x, a \right].$$

Here $\mathbb{E}_\pi[\cdot]$ denotes that all actions are chosen by policy $\pi$ and transitions are given by $\mathbb{P}$. The optimal policy $\pi^\star$ and optimal action-value function $Q^\star$ are defined recursively as

$$Q_h^\star(x, a) := r_h(x, a) + \mathbb{E}\left[ \max_{a'} Q^\star(x', a') \mid x, a \right], \qquad \pi_h^\star(x) = \operatorname*{argmax}_a Q_h^\star(x, a).$$

The optimal value function $V_h^\star$ is defined analogously.

The agent interacts with the MDP for $K$ episodes, where in episode $k$ the agent pick a policy $\pi_k$ and we generate the trajectory $\tau_k = (x_1^k, a_1^k, r_1^k, x_2^k, a_2^k, r_2^k \ldots, x_H^k, a_H^k, r_H^k)$ where (1) $x_1^k$ is chosen adversarially, (2) $a_h^k = \pi_k(x_h^k)$, (3) $x_{h+1}^k \sim \mathbb{P}(\cdot \mid x_h^k, a_h^k)$, (4) $r_h^k = r(x_h^k, a_h^k)$. We would like to choose actions to maximize the cumulative rewards $\sum_{h=1}^{H} r_h^k$.

Equipped with these definitions, we can state our performance criterion. Over the course of $K$ episodes, we would like to accumulate reward that is comparable to the optimal policy, formalized via the notion of regret:

$$\operatorname{Reg}(K) := \sum_{k=1}^{K} \left( V_1^\star(x_1^k) - \sum_{h=1}^{H} r_h^k \right).$$

In particular, we seek algorithms with regret rate that is sublinear in $K$. Note that we have not assumed that $|\mathcal{S}|$ and $|\mathcal{A}|$ are finite, and we also allow for the starting state $x_1^k$ to be chosen adversarially in each episode.

### 2.1   Metric spaces.

Instead of assuming that $|\mathcal{S}|$ and $|\mathcal{A}|$ are finite, we will posit a metric structure on these spaces. We recall the key definitions for metric spaces. A space $Y$ equipped with a function $\mathcal{D} : Y \times Y \to \mathbb{R}_+$ is a *metric space* if $\mathcal{D}$ satisfies (a) $\mathcal{D}(y, y') = 0$ iff $y = y'$ (b) $\mathcal{D}$ is symmetric, and (c) $\mathcal{D}$ satisfies the triangle inequality $\mathcal{D}(x, y) \leq \mathcal{D}(x, z) + \mathcal{D}(z, y)$. If these properties hold then $\mathcal{D}$ is called a *metric*. For a radius $r > 0$, we use the notation $B(y, r) := \{y' \in Y : \mathcal{D}(y, y') < r\}$ to denote the open ball centered at $y$ with radius $r$. For a subset $Y' \subseteq Y$ the *diameter* is defined as $\operatorname{diam}(Y') := \sup_{y, y' \in Y'} \mathcal{D}(y, y')$. We also use the standard notions of covering and packing to measure the size of metric spaces.

**Definition 1** (Notions of size). *A covering of $Y$ at scale $r$ (also called an $r$-covering) is a collection of subsets of $Y$, each with diameter at most $r$, whose union equals $Y$. The minimum number of subsets that form an $r$-covering is the $r$-covering number, denoted $N_r(Y)$. A packing of $Y$ at scale $r$ (also called an $r$-packing) is a collection of points $Z \subset Y$ such that $\min_{z \neq z' \in Z} D(z, z') \geq r$. The maximum number of points that form an $r$-packing is the $r$-packing number, denoted $N_r^{pack}(Y)$. An $r$-net of $Y$ is an $r$-packing $S \subset Y$ for which $\{B(y, r)\}_{y \in S}$ covers $Y$.*

These definitions also apply to subsets of the metric space, which will be important for our development. Also note that $N_{2r}^{\mathrm{pack}}(Y) \leq N_r(Y) \leq N_r^{\mathrm{pack}}(Y)$.

## 2.2 Main Assumptions.

We now state the main assumptions that we adopt in our analysis. These or closely related assumptions are standard in the literature on bandits and reinforcement learning in metric spaces (Song and Sun, 2019; Sinclair et al., 2019; Touati et al., 2020; Slivkins, 2014).

**Assumption 1.** *$(\mathcal{S} \times \mathcal{A}, \mathcal{D})$ is a metric space with finite diameter $\mathrm{diam}(\mathcal{S} \times \mathcal{A}) = d_{max} < \infty$.*

**Assumption 2.** *For every $h \in [H]$, $Q_h^\star$ is $L$-Lipschitz continuous with respect to $\mathcal{D}$:*

$$\forall (x, a), (x', a') : |Q_h^\star(x, a) - Q_h^\star(x', a')| \leq L \cdot \mathcal{D}((x, a), (x', a')). \tag{1}$$

*Additionally $V_h^\star$ is $L$-Lipschitz with respect to the metric $\mathcal{D}_X : (x, x') \mapsto \min_{a, a'} \mathcal{D}((x, a), (x', a'))$:*

$$\forall x, x' : |V_h^\star(x) - V_h^\star(x')| \leq L \cdot \min_{a, a'} \mathcal{D}((x, a), (x', a')). \tag{2}$$

Assumption 1 is a basic regularity condition, while the first part of Assumption 2 imposes continuity of the $Q^\star$ function. In particular, Lipschitz-continuity characterizes how the metric structure influences the reinforcement learning problem. These assumptions appear in prior work, and we note that (1) is strictly weaker than assuming that $\mathbb{P}$ is Lipschitz continuous (Kakade et al., 2003; Ortner and Ryabko, 2012).

The second part of Assumption 2 reflects an additional structural assumption on the problem, which is a departure from previous work. In detail, (2) posits that the optimal value function $V_h^\star$ is $L$-Lipschitz with respect to a metric defined only on the states that is derived from the original one. This metric is dominated by the original one since for each $(x, x', a)$ we have $\min_{a_1, a_2} \mathcal{D}((x, a_1), (x', a_2)) \leq \mathcal{D}((x, a), (x', a))$, so this assumption is not directly implied by (1). However, whenever $\mathcal{D}$ is sub-additive in the sense that $\mathcal{D}((x, a), (x', a')) \leq \mathcal{D}_S(x, x') + \mathcal{D}_A(a, a')$, then the assumption holds trivially. Sub-additivity holds for most metrics of interest, including those induced by $\ell_p$ norms for $p \geq 1$. As such, this assumption is not particularly restrictive.

## 2.3 Related work

Reinforcement learning in the tabular setting, where the state and action spaces are finite, is relatively well-understood (Azar et al., 2017; Dann et al., 2017; Zanette and Brunskill, 2019). Of this line of work, the two most related papers are those of of Jin et al. (2018) and Simchowitz and Jamieson (2019). Our results build on the model-free/martingale analysis of Jin et al. (2018), which has been used in recent work on RL in metric spaces (Song and Sun, 2019; Sinclair et al., 2019; Touati et al., 2020). We also employ techniques from the gap-dependent analysis of Simchowitz and Jamieson (2019). In particular, we use a version of their "clipping" argument, as we will explain in Section 5.

Moving beyond the tabular setting, several papers study reinforcement learning in metric spaces, originating with the results of Kakade et al. (2003) (c.f., Ortner and Ryabko (2012); Ortner (2013); Song and Sun (2019); Ni et al. (2019); Sinclair et al. (2019); Touati et al. (2020)). Of these, the most related result is that of Sinclair et al. (2019) who study the adaptive discretization algorithm and give a worst-case regret analysis, showing that the algorithm has a regret rate of $K^{\frac{d+1}{d+2}}$ where $d$ is the covering dimension of the metric space. Essentially the same results appear in Touati et al. (2020), although the algorithm is slightly different. However, none of these results give sharper instance-dependence guarantees that reflect benign problem structure, as we will obtain.

For the special case of (contextual) bandits, several instance-dependent guarantees that yield improved regret rates exist (Auer et al., 2007; Valko et al., 2013; Kleinberg et al., 2019; Bubeck et al., 2011;

Slivkins, 2014; Krishnamurthy et al., 2019). For non-contextual bandits, the results and assumptions vary considerably, but most results quantify a benign instance in terms of the size of the set of near-optimal actions. The formulation that we adopt is the notion of *zooming dimension*, which measures the growth rate of the $r$-packing number of the set of $O(r)$-suboptimal arms. This notion has been used in several works on bandits and contextual bandits in metric spaces, and we will recover some of these results as a special case of our main theorem.

## 3 Main Results

Our main result is a regret bound that scales with the *zooming dimension*. We introduce this parameter with a sequence of definitions. First, we define the gap function, which describes the sub-optimality of an action $a$ for state $x$.

**Definition 2** (Gap). *For any $(x, a) \in S \times A$, for $h \in [H]$, the stage-dependent sub-optimality gap is*

$$\text{gap}_h(x, a) := V_h^\star(x) - Q_h^\star(x, a).$$

We use the gaps to define the subset of the metric space that is near-optimal.

**Definition 3** (Near-optimal set). *We define near-optimal set as*

$$\mathcal{P}_{h,r}^{Q^\star} := \left\{ (x, a) \in \mathcal{S} \times \mathcal{A} : \text{gap}_h(x, a) \leq \left( \frac{2(H+1)}{d_{max}} + 2L \right) r \right\}.$$

Intuitively, $\mathcal{P}_{h,r}^{Q^\star}$ is the set of state-action pairs with gap that is $O(r)$ at stage $h$. The constant in the definition is a consequence of our analysis, but it is quite similar to the constant in the definition of Slivkins (2014) for contextual bandits. In particular, he considers $d_{\max} = 1, H = 1, L = 1$ and obtains a constant of 12, while we obtain a constant of 6 in this case.

Finally, we define the zooming number and the zooming dimension.

**Definition 4** (Zooming number and dimension). *The $r$-zooming number is the $r$-packing number of the near-optimal set $\mathcal{P}_{h,r}^{Q^\star}$, that is $N_r^{pack}(\mathcal{P}_{h,r}^{Q^\star})$. The stage-dependent zooming dimension is defined as*

$$z_{h,c} := \inf \left\{ d > 0 : N_r^{pack}(\mathcal{P}_{h,r}^{Q^\star}) \leq cr^{-d}, \forall r \in (0, d_{\max}] \right\}.$$

*The zooming dimension for the instance as the largest among all stages $z_c = \max_{h \in [H]} z_{h,c}$.*

Intuitively, the zooming dimension measures how the near-optimal region grows as we change the sub-optimality level $r$. Importantly, we use $r$ both to parametrize the radius in the packing number and the sub-optimality. Thus, the zooming number captures how many $r$-separated points can be packed into the $O(r)$ sub-optimal region.

The more standard notion of complexity of a metric space is the *covering dimension*, defined as

$$d_c := \inf\{d > 0, N_r^{\text{pack}}(\mathcal{S} \times \mathcal{A}) \leq cr^{-d}, \forall r \in (0, d_{\max}]\}.$$

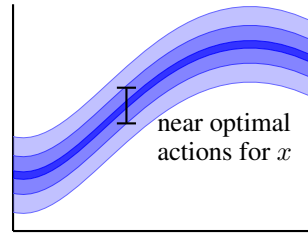

Figure 1: An example where the zooming dimension is 1 while the the covering dimension is 2.

Examining the definitions, it is clear that we have $z_c \leq d_c$, since the packing numbers are only smaller. However, in benign instances where the sub-optimal region concentrates to a low dimensional manifold, we may have $z_c < d_c$ (and possibly much smaller), which will enable sharper regret bounds. An example is illustrated in Figure 1, where the set of near-optimal actions concentrates on a narrow band for each $x$. Thus the entire space and hence the covering dimension is 2-dimensional, but the zooming dimension is 1. More generally, if $\mathcal{S}$ is a $d_S$ dimensional space and $\mathcal{A}$ is a $d_A$ dimensional space, then the covering dimension could be $\Omega(d_S + d_A)$ while the zooming dimension could be as small as $O(d_S)$.

With these definitions, we can now state the main theorem.

**Theorem 1.** *For any initial states $\{x_1^k : k \in [K]\}$, and any $\delta \in (0, 1)$, with probability at least, $1 - \delta$ Adaptive Q-learning has the following regret[2]*

$$\text{Reg}(K) \leq \tilde{O}\left(H^{3/2} \inf_{r_0 \in (0, d_{max}]} \left(\sum_{h=1}^{H} \sum_{r=d_{max}2^{-i}, r \geq r_0} N_r^{pack}(\mathcal{P}_{h,r}^{Q^\star})\frac{d_{max}}{r} + \frac{Kr_0}{d_{max}}\right)\right)$$
$$+ \tilde{O}\left(H^2 + \sqrt{H^3 K \log(1/\delta)}\right).$$

Before turning to a discussion of the theorem, we state some corollaries. First, by optimizing $r_0$, we obtain a regret bound in terms of the zooming dimension.

**Corollary 2.** *For any initial states $\{x_1^k : k \in [K]\}$, and any $\delta \in (0, 1)$, with probability at least $1 - \delta$ Adaptive Q-learning has $\text{Reg}(K) \leq \tilde{O}\left(H^{5/2} K^{\frac{z_c+1}{z_c+2}}\right)$, for any constant $c > 0$.*

Finally, we recover the regret rate of Slivkins (2014) in the special case of contextual bandits.

**Corollary 3** (Contextual bandits). *If $H = 1$, then Adaptive Q-learning has regret $\tilde{O}\left(K^{\frac{z_c+1}{z_c+2}}\right)$, which recovers the regret rate of Slivkins (2014).*

We now turn to the remarks:

- Theorem 1 gives a regret bound that depends on the packing numbers of the near-optimal set (Definition 3). This bound should be compared with the "metric-specific" regret guarantee of Sinclair et al. (2019) or the "refined regret bound" of Touati et al. (2020). Both of these results have the same form as ours with all terms in agreement, but with $N_r^{\text{pack}}(\mathcal{S} \times \mathcal{A})$ in the place of $N_r^{\text{pack}}(\mathcal{P}_{h,r}^{Q^\star})$. As $\mathcal{P}_{h,r}^{Q^\star} \subset \mathcal{S} \times \mathcal{A}$, our bound is always sharper.

- The more-interpretable bound is in terms of the zooming dimension (Definition 4), which highlights the dependence on the number of episodes $K$. We obtain a regret rate of $K^{\frac{z_c+1}{z_c+2}}$ for any constant $c > 0$, which should be compared with the non-adaptive rate $K^{\frac{d_c+1}{d_c+2}}$ that scales with the covering dimension (Song and Sun, 2019; Sinclair et al., 2019; Touati et al., 2020).[3] As the zooming dimension can be smaller than covering dimension (recall Figure 1), this bound demonstrates a polynomial improvement over non-adaptive approaches.

- Corollary 3 shows that our bound recovers the guarantee from Slivkins (2014), although his bound does not require that (2) holds. We give a more detailed explanation on the necessity of (2) in Section 5. Nevertheless, the fact that we essentially recover his bound suggests that our results are the natural generalization to multi-step RL.

- Finally, we remark that we can instantiate the result in the tabular setting with finite $\mathcal{S}, \mathcal{A}$ by taking the metric to be $\mathcal{D}((x, a), (x', a')) = \mathbf{1}\{(x, a) \neq (x', a')\}$. In this case we obtain a "partial" gap-dependent bound of the form:

$$\text{poly}(H) \cdot \left(\sqrt{|\mathcal{S}|K} + \sum_{h=1}^{H} \sum_{x \in \mathcal{S}} \sum_{a:\text{gap}_h(x,a)>0} \frac{\log(K)}{\text{gap}_h(x, a)}\right).$$

This is not a fully gap-dependent bound because of the $\sqrt{|\mathcal{S}|K}$ term, but it does recover an intermediate result of Simchowitz and Jamieson (2019). In particular, this confirms that the model-free methods can achieve a partial gap-dependent guarantee for the tabular setting.

## 4 Algorithm

As we have mentioned, the algorithm we analyze is the Adaptive $Q$-learning algorithm of Sinclair et al. (2019). For completeness, the pseudocode is reproduced in Algorithm 1. The algorithm adaptively partitions the state-action space to focus on the informative regions, and it uses optimism to explore the space and drive the agent to regions with high reward.

**Algorithm 1** Adaptive $Q$–learning

---

1: For $h \in [H]$, initialize $\mathcal{P}_h^1$ to be a single ball $B_h$ with radius $d_{max}$. $Q_h^1(B_h) \leftarrow H$.
2: **for** each episode $k = 1, 2, \ldots, K$ **do**
3:      Receive $x_1^k$.
4:      **for** stage $h = 1, 2, \ldots, H$ **do**
5:          $B_h^k = \text{argmax}_{B \in \text{rel}_h^k(x_h^k)} Q_h^k(B)$
6:          Play action $a_h^k$ for some $(x_h^k, a_h^k) \in \text{dom}_h^k(B_h^k)$
7:          Receive $r_h^k, x_{h+1}^k$, update $t = n_h^{k+1}(B_h^k) = n_h^k(B_h^k) + 1$
8:          $V_{h+1}^k(x_{h+1}^k) = \min\left\{H, \max_{B \in \text{rel}_{h+1}^k(x_{h+1}^k)} Q_{h+1}^k(B)\right\}$.
9:          $Q_h^{k+1}(B_h^k) = (1 - \alpha_t)Q_h^k(B_h^k) + \alpha_t(r_h^k + b_t + V_{h+1}^k(x_{h+1}^k))$.
10:          **if** $t_h^{k+1}(B_h^k) \geq \left(\frac{d_{max}}{r(B_h^k)}\right)^2$ **then** split $B_h^k$:
11:              Create a set of balls $\mathcal{B}_h^k = \{\frac{1}{2}r(B_h^k)\text{-net of } \text{dom}_h^k(B_h^k)\}$.
12:              Inherit the count and $Q_h^k$ from $B_h^k$. Set $\mathcal{P}_h^{k+1} = \mathcal{P}_h^k \cup \mathcal{B}_h^k$.
13:          **end if**
14:      **end for**
15: **end for**

---

During the execution, the algorithm creates many balls $B \subset \mathcal{S} \times \mathcal{A}$ for each stage $h$. We use $\mathcal{P}_h^k$ to denote the set of balls created for stage $h$ up until episode $k$. Every ball $B$ has a radius, denoted $r(B)$ and a *domain*, denoted $\text{dom}_h^k(B)$. The domain is the set of points contained in this ball, but not in any other active ball with smaller radius. Formally,

$$\text{dom}_h^k(B) := B \setminus \{\cup_{B' \in \mathcal{P}_h^k : r(B') < r(B)} B'\}.$$

For each ball, we also maintain a counter $t = n_h^k(B)$ which denotes the number of times we have chosen state-action pairs in $B$ or its ancestors. Parents and ancestors are defined via the splitting rule: when a ball is split in line 10, the resulting balls are called the children. Finally, we maintain a scalar $Q_h^k(B)$ which serves as an upper bound on $\max_{(x,a) \in B} Q_h^\star(x, a)$.

In stage $h$ of episode $k$, we select the action for state $x_h^k$ as follows: we consider all the smallest balls that contains $x_h^k$, defined as "relevant" balls

$$\text{rel}_h^k(x) := \{B | \exists a, (x, a) \in \text{dom}_h^k(B)\}.$$

Among the relevant balls, the algorithm select the ball $B_h^k$ with the highest $Q_h^k(B)$ value and plays an arbitrary action such that $(x_h^k, a) \in B_h^k$. We increment the sample count $n_h^k(B_h^k)$ for this ball and at the end of the episode, we update $Q_h^k(B_h^k)$ via

$$Q_h^{k+1}(B_h^k) = (1 - \alpha_t)Q_h^k(B_h^k) + \alpha_t(r_h^k + b_t + V_{h+1}^k(x_{h+1}^k))$$

$$V_{h+1}^k(x) = \min\left\{H, \max_{B \in \text{rel}_{h+1}^k(x)} Q_{h+1}^k(B)\right\}.$$

where the $\alpha_t$ is the learning rate and $b(t)$ is the bonus added to ensure $Q_h^k$ is optimistic. Formally,

$$\alpha_t := \frac{H+1}{H+t}, \qquad b_t := 2\sqrt{\frac{H^3 \log(4HK/\delta)}{t}} + \frac{4Ld_{max}}{\sqrt{t}}.$$

For all other balls at stage $h$, we set $Q_h^{k+1}(B) \leftarrow Q_h^k(B)$, with no update.

We split a ball $B$ as soon as $n_h^k(B) \geq \left(\frac{d_{max}}{r(B)}\right)^2$. When splitting, we create a set of new "children" balls with radius $r(B)/2$ that forms an $r(B)/2$-net of $\text{dom}_h^k(B)$. These balls inherit the count $n_h^k$ and the estimate $Q_h^k$ from the "parent" ball $B$, and we add them to $\mathcal{P}_h^{k+1}$. This splitting rule leads to the following invariant

**Lemma 4** (Lemma 5.3 in Sinclair et al. (2019)). *For every $(h, k) \in [H] \times [K]$, we have*

     1. *(Covering) The domains of balls in $\mathcal{P}_h^k$ covers $\mathcal{S} \times \mathcal{A}$.*
     2. *(Separation) For any two balls of radius $r$, their centers are at distance at least $r$.*

**Computational considerations.** As discussed in Sinclair et al. (2019), this algorithm can be implemented in a computationally efficient manner provided that the metric space allows certain natural operations. Formally, we operate in an oracle model, which allows us to query the metric to compute $\mathrm{dom}(B), \mathrm{rel}(x)$, and to construct an $r$-net for any $r$ and any subset of the metric space.

## 5 Proof sketch

In this section we describe the main steps of the proof, with details deferred to the appendix.

It is worth reviewing prior regret analyses for episodic RL (Jin et al., 2018). The arguments establish a regret decomposition that relates the estimate $V_1^k$ to $V^{\pi_k}$, the expected reward collected in episode $k$. The decomposition is recursive in nature, involving differences between $Q_h^k$ and $Q_h^\star$. These are controlled by the update rule and the design of the learning rate. In particular, we can bound $Q_h^k - Q_h^\star$ by an immediate "surplus" $\beta_t$ and the downstream value function error. Formally for any ball $B$ with $(x,a) \in \mathrm{dom}_h^k(B)$

$$Q_h^k(B) - Q_h^\star(x,a) \le \mathbf{1}_{[t=0]} H + \sum_{i=1}^t \alpha_t^i (V_{h+1}^{k_i} - V_{h+1}^\star)(x_{h+1}^{k_i}) + \beta_t, \qquad (3)$$

where $t = n_h^k(B), \alpha_i^t = \alpha_i \prod_{j=i+1}^t (1 - \alpha_j)$ and $\beta_t = 2 \sum_{i=1}^t \alpha_i^t b_i$. Here $k_i$ is the index of the episode where $B$ was selected for the $i^{\mathrm{th}}$ time. Summing over all episodes and grouping terms appropriately, we obtain

$$\sum_{k=1}^K (V_h^k - V_h^{\pi_k})(x_h^k) \le \sum_{k=1}^K \left( H \mathbf{1}_{[n_h^k = 0]} + \beta_{n_h^k} + \xi_h^k \right) + (1 + {}^1\!/_H) \sum_{k=1}^K \left( V_{h+1}^k - V_{h+1}^{\pi_k} \right)(x_{h+1}^k),$$

where $\xi_{h+1}^k$ is a stochastic term that can be ignored for this discussion. Note that, as long as $V_h^k$ is optimistic (which we will verify), this also provides a bound on the regret.

For the tabular setting, Jin et al. (2018) use this regret decomposition to obtain a worst-case bound. The leading term arises from the "surplus" term $\beta_{n_h^k}$, which leads to a $\mathrm{poly}(H)\sqrt{SAK}$ regret bound for the tabular setting. On the other hand for our setting, the splitting rule implies that for any ball $B$, we must have $n_h^k \le \left( {}^{d_{\max}}\!/_{r(B)} \right)^2$. We can use this to obtain a bound that depends on the number of active balls at each scale $r$ times $d_{\max}/r$. If we could bound the number of active balls at scale $r$ in terms of the packing number $N_r^{\mathrm{pack}}(\mathcal{P}_{h,r}^{Q^\star})$, then we would obtain the instance-dependent bound.

Unfortunately, this is not possible. In general, the algorithm will activate balls outside of the near-optimal region, because we may have to select a highly suboptimal ball many times to reduce downstream over-estimation error. So indeed the number of active balls at scale $r$ could be much larger than the packing of the near-optimal set.

We address this with the following key observation. If the surplus $\beta_{n_h^k}$ is small compared to gap, and we choose this ball, it must be the case that the downstream regret is quite large, otherwise we would not have chosen this ball. If this is true, we can account for the surplus by adding a small constant fraction of the future regret. In otherwords, we can "clip" the surplus to zero once it is proportional to the gap, and we only pay a constant factor in the recursive term. This is the clipping trick developed by Simchowitz and Jamieson (2019) to establish gap dependent bounds for tabular MDP. Formally instead of (3), we have the following lemma.

**Lemma 5** (Clipped upper bound). *For any $\delta \in (0,1)$ with probability at least $1 - \delta/2, \forall h \in [H]$,*

$$Q_h^k(B_h^k) - Q_h^\star(x_h^k, a_h^k) \le (1 + {}^1\!/_H) \left( \mathbf{1}_{[t=0]} H + \sum_{i=1}^t \alpha_t^i (V_{h+1}^{k_i} - V_{h+1}^\star)(x_{h+1}^{k_i}) \right)$$

$$+ \mathrm{clip}\left[ \beta_t \mid \frac{\mathrm{gap}_h(x_h^k, a_h^k)}{H+1} \right],$$

*where $t = n_h^k(B), \alpha_i^t = \alpha_i \prod_{j=i+1}^t (1 - \alpha_j)$ and $\beta_t = 2 \sum_{i=1}^t \alpha_i^t b_i$ and $\mathrm{clip}[\mu \mid \nu] := \mu \mathbf{1}\{\mu \ge \nu\}$.*

This bound should be compared with (3). On one hand the recursive term is multiplied by $1 + {}^1\!/_H$, but, on the other, we are able to clip the surpluses $\beta_t$. The former will exponentiate but will asymptote to $e$, while the latter is crucial for our instance dependent bounds.

Using this lemma, we can bound the difference between $V_h^k$ and $V_h^{\pi^k}$.

**Lemma 6** (Clipped recursion, informal). *For any $\delta \in (0, 1)$, with probability at least $1 - \delta/2$, $\forall h \in [H]$,*

$$\sum_{k=1}^{K}(V_h^k - V_h^{\pi^k})(x_h^k) \le \sum_{k=1}^{K}(1 + 1/H)\,\alpha_t^0 + \text{clip}\left[\beta_{n_h^k} \mid \text{gap}_h(x_h^k, a_h^k)/(H+1)\right] + \xi_{h+1}^k$$

$$+ (1 + 1/H)^2 \sum_{k=1}^{K}(V_{h+1}^k - V_{h+1}^{\pi^k})(x_{h+1}^k),$$

*where $\xi_{h+1}^k$ is conditionally centered random variable with range $H$.*

We bound $V_1^k - V_1^{\pi^k}$, and by optimism the regret, by applying Lemma 6 recursively.

The last step is to show that the sum of clipped surpluses can be related to the zooming dimension. First note that for any ball $B$, its ancestors must be played at least $1/4\left(d_{\max}/r(B)\right)^2$ times before it becomes activated. Since the ball inherits data from its ancestors, if it becomes activated but only contains points with large gap, we can always clip the surplus term. Thus all active balls $B$ that have $r(B) \ll \min_{x,a \in B} \text{gap}(x, a)$ do not contribute to the regret.

Next, if a ball with radius $r$ contains a point where the gap is small, we cannot appeal to clipping. However, by Lipschitzness, all points in the ball must have small gaps, which means that this ball is contained in the near optimal set at scale $r$. As above, the surplus for each of these balls contributes at most $d_{\max}/r$ to the regret. Then, since all balls with radius $r$ are at least $r$ apart and we only incur regret for those entirely contained in the near-optimal region, we obtain the bound that depends on $N_r^{\text{pack}}(\mathcal{P}_{h,r}^{Q^\star})$.

**Remarks on Assumption 2.** We give some intuition on why our proof requires (2), which is slightly stronger than what is required for the zooming dimension analysis of Slivkins (2014) for contextual bandits. In Slivkins (2014), the optimistic selection rule ensures that the context-action pairs chosen by the algorithm have small gap, but this is not true in the multi-step setting. In the RL setting, we might select an action (in a ball) with a large gap because the downstream regret is large. In this case, we can clip the surplus, but we can only clip at the *minimum* gap among all $(x, a)$ pairs in the ball. To obtain a zooming dimension bound, we must argue that this ball is contained in the near-optimal set, but this requires that the value functions, and hence the gaps, are Lipschitz. We recall that (2) is implied by (1) if the metric is sub-additive.

## 6 Discussion

In this paper, we give a refined analysis of the Adaptive Q-learning algorithm of Sinclair, Banerjee and Yu (2019) for sample efficient reinforcement learning in metric spaces. We show that the algorithm has a regret bound that depends on the zooming dimension of the instance, with rate $K^{\frac{z+1}{z+2}}$ when the zooming dimension is $z$. This always improves on the worst-case bound that depends on the covering dimension, and can be much better when the $Q^\star$ function concentrates quickly onto a low-dimensional set of actions. The bound also recovers that of Slivkins (2014) for contextual bandits in metric spaces, under a slightly stronger assumption. The key technique is the clipped regret decomposition of Simchowitz and Jamieson (2019), which we complement with a book-keeping argument. Our results show that adaptivity to benign instances is possible in RL with metric spaces, and partially mitigate the curse of dimensionality in such settings.

## Broader Impact

This is primarily a theoretical contribution, so the broader impacts of the results are quite minimal. As the algorithm we analyze has already been developed and used in experiments (Sinclair et al., 2019), we do not expect our results would have any bearing on whether this algorithm is used in practice. On the other hand, we do believe that adaptive algorithms for reinforcement learning will be essential for applications where data collection is costly, such as applications in tutoring systems. We hope that our results can inspire future work into these important applications.

## Acknowledgments and Disclosure of Funding

We thank Wen Sun and Aleksandrs Slivkins for formative discussions during the conception of this paper. We also thank Max Simchowitz for insightful discussions regarding the clipping technique. TC is supported by a scholarship from Deeproute.ai Ltd and is an employee of Deeproute.ai Ltd. AK gratefully acknowledges support from Microsoft Corporation and NSF Award IIS-1763618. AK is also affiliated with the University of Massachusetts Amherst.

## Footnotes

[1]Deterministic rewards simplifies the presentation but has no bearing on the final results. In particular, we can handle stochastic bounded rewards with minimal modification to the proofs.

[2]Throughout the paper $\tilde{O}(\cdot)$ suppresses logarithmic dependence in its argument.

[3]We always treat $c$ as a universal constant, so its dependence in the regret bounds is suppressed.

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
