[Supplementary Material]

# 7 Appendix

In this section we provide a detailed proof for the main theorem. First we state some facts about the learning rate and the algorithm.

**Lemma 7** (Lemma 4.1 from Jin et al. (2018)). *Let $\alpha_t^i := \alpha_i \prod_{j=i+1}^t (1 - \alpha_j)$. Then for every $i \geq 1$:*

$$\sum_{t=i}^{\infty} \alpha_t^i = 1 + \frac{1}{H}.$$

**Lemma 8** (Lemma 5.4 from Sinclair et al. (2019)). *For any $h \in [H]$ and ball $B \in \mathcal{P}_h^K$ the number of time $B$ is selected is bounded by*

$$|\{k : B_h^k = B\}| \leq \frac{3}{4} \left( \frac{d_{max}}{r(B)} \right)^2$$

*Moreover, the number of times that ball $B$ and its ancestors have been played is at least $\frac{1}{4} \left( \frac{d_{max}}{r(B)} \right)^2$.*

To bound the regret, our starting point is an upper bound on the difference between the optimistic $Q$–function and the optimal $Q^\star$ function.

**Lemma 9** (Lemma E.7 from Sinclair et al. (2019)). *For any $\delta \in (0,1)$ if $\beta_t = 2 \sum_{i=1}^t \alpha_t^i b(i)$ then*

$$\beta_t \leq 8 \sqrt{\frac{H^3 \log(4HK/\delta)}{t}} + 16 \frac{L d_{max}}{\sqrt{t}}$$

*With probability at least $1 - \delta/2$ the following holds simultaneously for all $(x, a, h, k) \in \mathcal{S} \times \mathcal{A} \times [H] \times [K]$ and ball $B$ such that $(x,a) \in \mathrm{dom}_h^k(B)$. $t = n_h^k(B)$ and $k_1 < \cdots < k_t$ are the episodes where $B$ or its ancestors were encountered previously by the algorithm.*

$$0 \leq Q_h^k(B) - Q_h^\star(x,a) \leq \mathbf{1}_{[t=0]} H + \beta_t + \sum_{i=1}^t \alpha_t^i (V_{h+1}^{k_i} - V_{h+1}^\star)(x_{h+1}^{k_i})$$

This bound contains three parts. The first is an upper bound for the first step when there is no data. The second term, $\beta_t$, is the surplus that we add to be optimistic. The third part is an "average" of the estimated future regret. The key observation is that when $\beta_t$ is small, it can be absorbed into the future surplus. So we can clip $\beta_t$ proportional to the future regret, or gap. This enables a gap dependent regret bound.

**Lemma 10** (Clipped upper bound). *For any $\delta \in (0,1)$ if $\beta_t = 2 \sum_{i=1}^t \alpha_t^i b(i)$. With probability at least $1 - \delta/2$, $\forall h \in [H], k \in [K]$,*

$$Q_h^k(B_h^k) - Q_h^\star(x_h^k, a_h^k) \leq \left( 1 + \frac{1}{H} \right) \left( \mathbf{1}_{[t=0]} H + \sum_{i=1}^t \alpha_t^i (V_{h+1}^{k_i} - V_{h+1}^\star)(x_{h+1}^{k_i}) \right)$$
$$+ \mathrm{clip} \left[ \beta_t \mid \mathrm{gap}_h(x_h^k, a_h^k)/(H+1) \right]$$

*Proof.* We use $a_h^\star : \mathcal{X} \to \mathcal{A}$ to denote a mapping from the state to the optimal action at stage $h$. By the definition of the gap

$$\mathrm{gap}_h(x_h^k, a_h^k) = Q^\star(x_h^k, a_h^\star(x_h^k)) - Q^\star(x_h^k, a_h^k) \leq Q_h^k(B_h^{k\star}) - Q_h^\star(x_h^k, a_h^k)$$

$$\leq Q_h^k(B_h^k) - Q_h^\star(x_h^k, a_h^k) \leq \mathbf{1}_{[t=0]} H + \beta_t + \sum_{i=1}^t \alpha_t^i (V_{h+1}^{k_i} - V_{h+1}^\star)(x_{h+1}^{k_i}),$$

where $B_h^{k\star}$ is the smallest ball that contains $(x_h^k, a_h^\star(x_h^k))$. The first inequality is by the lower bound of Lemma 9. Note that $B_h^{k\star} \in \mathrm{dom}_h^k(x_h^k)$. The second uses the selection rule of choosing the ball with the largest $Q_h^k(B)$ for $B \in \mathrm{dom}_h^k(x_h^k)$. The third inequality is by the upper bound of Lemma 9.

Now we consider two cases, if $\beta_t > \mathrm{gap}_h(x_h^k, a_h^k)/(H+1)$, the bound is trivially implied by Lemma 9. If $\beta_t \leq \mathrm{gap}_h(x_h^k, a_h^k)/(H+1)$,

$$\mathrm{gap}_h(x_h^k, a_h^k) \leq \mathbf{1}_{[t=0]}H + \beta_t + \sum_{i=1}^t \alpha_t^i(V_{h+1}^{k_i} - V_{h+1}^\star)(x_{h+1}^{k_i})$$

$$\leq \mathbf{1}_{[t=0]}H + \sum_{i=1}^t \alpha_t^i(V_{h+1}^{k_i} - V_{h+1}^\star)(x_{h+1}^{k_i}) + \mathrm{gap}_h(x_h^k, a_h^k)/(H+1)$$

Taking the gap to one side we have

$$\mathrm{gap}_h(x_h^k, a_h^k) \leq \frac{H+1}{H}\left(\mathbf{1}_{[t=0]}H + \sum_{i=1}^t \alpha_t^i(V_{h+1}^{k_i} - V_{h+1}^\star)(x_{h+1}^{k_i})\right)$$

By Lemma 9 and our assumption

$$Q_h^k(B_h^k) - Q_h^\star(x_h^k, a_h^k) \leq \mathbf{1}_{[t=0]}H + \beta_t + \sum_{i=1}^t \alpha_t^i(V_{h+1}^{k_i} - V_{h+1}^\star)(x_{h+1}^{k_i})$$

$$< \mathbf{1}_{[t=0]}H + \mathrm{gap}_h(x_h^k, a_h^k)/(H+1) + \sum_{i=1}^t \alpha_t^i(V_{h+1}^{k_i} - V_{h+1}^\star)(x_{h+1}^{k_i})$$

$$\leq \left(1 + \frac{1}{H}\right)\left(\mathbf{1}_{[t=0]}H + \sum_{i=1}^t \alpha_t^i(V_{h+1}^{k_i} - V_{h+1}^\star)(x_{h+1}^{k_i})\right). \qquad \square$$

The next step is to replace the future regret to $V^\star$ with the future regret of $V^{\pi_k}$, so that we can solve for the $h = 1$ case recursively.

**Lemma 11** (Clipped recursion). *For any $\delta \in (0,1)$ if $\beta_t = 2\sum_{i=1}^t \alpha_t^i b(i)$. With probability at least $1 - \delta/2$, $\forall h \in [H], k \in [K]$,*

$$\sum_{k=1}^K (V_h^k - V_h^{\pi^k})(x_h^k) \leq \sum_{k=1}^K \left(1 + \frac{1}{H}\right)\left(H\mathbf{1}_{[n_h^k=0]} + \xi_{h+1}^k + \mathrm{clip}\left[\beta_{n_h^k} \mid \frac{\mathrm{gap}_h(x_h^k, a_h^k)}{H+1}\right]\right)$$

$$+ \left(1 + \frac{1}{H}\right)^2 \sum_{k=1}^K (V_{h+1}^k - V_{h+1}^{\pi^k})(x_{h+1}^k),$$

*where $\xi_{h+1}^k = \mathbb{E}\left[V_{h+1}^\star(x) - V_{h+1}^{\pi_k}(x) \mid x_h^k, a_h^k\right] - (V_{h+1}^\star - V_{h+1}^{\pi_k})(x_{h+1}^k)$.*

*Proof.*

$$V_h^k(x_h^k) - V_h^{\pi^k}(x_h^k) \leq \max_{B \in \mathrm{rel}_h^k(x_h^k)} Q_h^k(B) - Q_h^{\pi^k}(x_h^k, a_x^k) = Q_h^k(B_h^k) - Q_h^{\pi^k}(x_h^k, a_x^k)$$

$$= Q_h^k(B_h^k) - Q_h^\star(x_h^k, a_h^k) + Q_h^\star(x_h^k, a_h^k) - Q_h^{\pi^k}(x_h^k, a_x^k)$$

$$= \left(1 + \frac{1}{H}\right)\left(\mathbf{1}_{[t=0]}H + \sum_{i=1}^t \alpha_t^i(V_{h+1}^{k_i} - V_{h+1}^\star)(x_{h+1}^{k_i})\right) + \mathrm{clip}\left[\beta_t \mid \frac{\mathrm{gap}_h(x_h^k, a_h^k)}{H+1}\right]$$

$$+ (V_{h+1}^\star - V_{h+1}^{\pi^k})(x_{h+1}^k) + \xi_{h+1}^k.$$

Summing over the episodes, let $n_h^k = n_h^k(B_h^k)$ and $k_i(B_h^k)$ be the episode where $B_h^k$ or its ancestors are sampled for the $i$-th time.

$$\sum_{k=1}^K V_h^k(x_h^k) - V_h^{\pi^k}(x_h^k) \leq \sum_{k=1}^K \left(1 + \frac{1}{H}\right)\left(\mathbf{1}_{[t=0]}H + \mathrm{clip}[\beta_t, \frac{\mathrm{gap}_h(x_h^k, a_h^k)}{H+1}]\right)$$

$$+ \left(1 + \frac{1}{H}\right)\sum_{k=1}^K \sum_{i=1}^{n_h^k} \alpha_{n_h^k}^i(V_{h+1}^{k_i(B_h^k)} - V_{h+1}^\star)(x_{h+1}^{k_i(B_h^k)})$$

$$+ \sum_{k=1}^K \left((V_{h+1}^\star - V_{h+1}^{\pi^k})(x_{h+1}^k) + \xi_{h+1}^k\right).$$

Using the observation in Jin et al. (2018); Song and Sun (2019), for the second term we can rearrange the sum and use Lemma 7

$$\sum_{k=1}^{K}\sum_{i=1}^{n_h^k}\alpha_{n_h^k}^i(V_{h+1}^{k_i(B_h^k)}-V_{h+1}^\star)(x_{h+1}^{k_i(B_h^k)})\le \sum_{k=1}^{K}(V_{h+1}^k-V_{h+1}^\star)(x_{h+1}^k)\sum_{t=n_h^k}^{\infty}\alpha_t^{n_h^k}$$

$$\le \left(1+\frac{1}{H}\right)\sum_{k=1}^{K}(V_{h+1}^k-V_{h+1}^\star)(x_{h+1}^k).$$

Since $V_{h+1}^{\pi^k}(x_{h+1}^k)\le V_{h+1}^\star(x_{h+1}^k)$, we have

$$\left(1+\frac{1}{H}\right)^2\sum_{k=1}^{K}(V_{h+1}^k-V_{h+1}^\star)(x_{h+1}^k)+\sum_{k=1}^{K}(V_{h+1}^\star-V_{h+1}^{\pi^k})(x_{h+1}^k)$$

$$\le \left(1+\frac{1}{H}\right)^2\left(\sum_{k=1}^{K}(V_{h+1}^k-V_{h+1}^\star)(x_{h+1}^k)+\sum_{k=1}^{K}(V_{h+1}^\star-V_{h+1}^{\pi^k})(x_{h+1}^k)\right)$$

$$= \left(1+\frac{1}{H}\right)^2\sum_{k=1}^{K}(V_{h+1}^k-V_{h+1}^{\pi^k})(x_{h+1}^k)$$

So we have

$$\sum_{k=1}^{K}(V_h^k-V_h^{\pi^k})(x_h^k)\le \sum_{k=1}^{K}\left(1+\frac{1}{H}\right)\left(H\mathbf{1}_{[n_h^k=0]}+\xi_{h+1}^k+\mathrm{clip}\left[\beta_{n_h^k}\mid\frac{\mathrm{gap}_h(x_h^k,a_h^k)}{H+1}\right]\right)$$

$$+\left(1+\frac{1}{H}\right)^2\sum_{k=1}^{K}(V_{h+1}^k-V_{h+1}^{\pi^k})(x_{h+1}^k). \qquad \square$$

There are two terms that we need to bound. The $\xi_{h+1}^k$ term can be bounded by a concentration argument as shown in Sinclair et al. (2019).

**Lemma 12** (Azuma–Hoeffding bound, Lemma E.9 from Sinclair et al. (2019)). *For any $\delta\in(0,1)$, with probability at least $1-\delta/2$*

$$\sum_{h=1}^{H}\sum_{k=1}^{k}\xi_{h+1}^k\le 2\sqrt{2H^3K\log(4HK/\delta)}$$

The clipped $\beta_t$ term requires a more refined treatment to relate it to the zooming number or zooming dimension. Recall our definition of the near-optimal space

$$\mathcal{P}_{h,r}^{Q^\star}=\{(x,a):\mathrm{gap}_h(x,a)\le c_1 r\},$$

where $c_1=\frac{2(H+1)}{d_{\max}}+2L$. Define the stage-dependent zooming number as

$$z_{h,c}=\inf\{d>0:|\mathcal{P}_{h,r}^{Q^\star}|\le cr^{-d}\}.$$

The following is our key lemma that bounds surplus $\beta_t$ using the zooming number.

**Lemma 13.**

$$\sum_{h=1}^{H}\sum_{k=1}^{K}\mathrm{clip}[\beta_{n_h^k},\frac{\mathrm{gap}_h(x_h^k,a_h^k)}{H+1}]\le \sum_{h=1}^{H}32(\sqrt{H^3\log(4HK/\delta)}+Ld_{\max})$$

$$\inf_{r_0\in(0,d_{\max}]}\left(\sum_{r=d_{\max}2^{-i},r\ge r_0}N_r^{pack}(\mathcal{P}_{h,r}^{Q^\star})\frac{d_{\max}}{r}+\frac{Kr_0}{d_{\max}}\right)$$

*Proof.* Let $c_2 = 16(\sqrt{H^3 \log(4HK/\delta)} + Ld_{\max})$. By Lemma 9 we have

$$\beta_{n_h^k} \leq 16(\sqrt{H^3 \log(4HK/\delta)} + Ld_{\max}) \frac{1}{\sqrt{n_h^k}} = c_2 \frac{1}{\sqrt{n_h^k}}$$

Let $n_{\min}(B) = \frac{1}{4} \left( \frac{d_{\max}}{r(B)} \right)^2$, and $n_{\max}(B) = \left( \frac{d_{\max}}{r(B)} \right)^2$. Considering Lemma 8 and the fact that a ball inherits samples from its parent, we know that for all $h$ and $k$,

$$n_{\min}(B) \leq n_h^k(B) \leq n_{\max}(B)$$

We rearrange the sum for each ball.

$$\sum_{k=1}^K \mathrm{clip}\left[ \beta_{n_h^k} \mid \frac{\mathrm{gap}_h(x_h^k, a_h^k)}{H+1} \right] \leq \sum_{B \in \mathcal{P}_h^K} \sum_{n=n_{\min}(B)}^{n_{\max}(B)} \mathrm{clip}\left[ c_2 \frac{1}{\sqrt{n}} \mid \frac{\mathrm{gap}_h(B)}{H+1} \right]$$

$$\leq c_2 \sum_{B \in \mathcal{P}_h^K} \sum_{n=n_{\min}(B)}^{n_{\max}(B)} \mathrm{clip}\left[ \frac{1}{\sqrt{n}}, \frac{\mathrm{gap}_h(B)}{H+1} \right]$$

The last step is due to the fact that $c_2 > 1$ and if $\frac{c_2}{\sqrt{n}} < \frac{\mathrm{gap}_h(B)}{H+1}$ then $\frac{1}{\sqrt{n}} < \frac{\mathrm{gap}_h(B)}{H+1}$. Now, ignoring clipping, the inner sum can be bounded by

$$\sum_{n=n_{\min}(B)}^{n_{\max}(B)} \frac{1}{\sqrt{n}} \leq \int_{i=1}^{\frac{3}{4} \left( \frac{d_{\max}}{r(B)} \right)^2} \frac{1}{\sqrt{i + \frac{1}{4} \left( \frac{d_{\max}}{r(B)} \right)^2}} \leq 2 \frac{d_{\max}}{r(B)}.$$

For clipping, let $\mathrm{gap}_h(B) = \min_{(x,a) \in B} \mathrm{gap}_h(x, a)$ be the gap for a ball $B$. We consider two cases.
**Case 1:** $\mathrm{gap}_h(B) \geq \frac{2(H+1)r(B)}{d_{\max}}$, we have

$$\frac{1}{\sqrt{n_h^k(B)}} \leq \frac{1}{\sqrt{n_{\min}(B)}} = \frac{2r(B)}{d_{\max}} \leq \frac{\mathrm{gap}_h(B)}{H+1}$$

So in this case the regret on ball $B$ is always clipped.

**Case 2:** $\mathrm{gap}_h(B) < \frac{2(H+1)r(B)}{d_{\max}}$

Let $(x_c, a_c)$ be the center of $B$ and $(x_m, a_m) \in B$ be the point that has the minimum gap, i.e. the point that achieves $\mathrm{gap}_h(B)$. Using the assumption that $Q^\star$ and $V^\star$ are Lipschitz:

$$\mathrm{gap}_h(x_c, a_c) - \mathrm{gap}_h(B) = Q_h^\star(x_c, a_h^\star(x_c)) - Q_h^\star(x_c, a_c) - (Q_h^\star(x_m, a_h^\star(x_m)) - Q_h^\star(x_m, a_m))$$
$$\leq 2Lr(B)$$

So we know that all the points in $B$ have small gaps relative to $r$.

$$\mathrm{gap}_h(x_c, a_c) \leq \mathrm{gap}_h(B) + 2Lr(B) \leq \frac{2(H+1)r(B)}{d_{\max}} + 2Lr(B).$$

Thus, we have $(x_c, a_c) \in \mathcal{P}_{h,r(B)}^{Q^\star}$. Now we are ready bound the sum. Note that for a ball $B \in \mathcal{P}_h^K$, either $B$ gets clipped, or the center of $B$ is in $\mathcal{P}_{h,r(B)}^{Q^\star}$. Since all the balls of radius $r$ are at least $r$ apart, we can have at most $N_r^{\mathrm{pack}}(\mathcal{P}_{h,r}^{Q^\star})$ in the latter case.

$$\sum_{k=1}^K \mathrm{clip}\left[ \beta_{n_h^k} \mid \frac{\mathrm{gap}_h(x_h^k, a_h^k)}{H+1} \right] \leq \sum_{B \in \mathcal{P}_h^K} \sum_{n=n_{\min}(B)}^{n_{\max}(B)} \mathrm{clip}[c_2 \frac{1}{\sqrt{n}} \mid \frac{\mathrm{gap}_h(B)}{H+1}]$$

$$\leq c_2 \inf_{r_0 \in (0, d_{\max}]} \left( \sum_{r=d_{\max}2^{-i}, r \geq r_0} N_r^{\mathrm{pack}}(\mathcal{P}_{h,r}^{Q^\star}) \frac{2d_{\max}}{r} + \frac{2Kr_0}{d_{\max}} \right).$$

The second term uses the fact that for any ball $B$ with $r(B) \leq r_0$, we have $n_{\min} \leq \frac{1}{4} \left( \frac{d_{\max}}{r_0} \right)^2$. $\quad \square$

Now we are ready to prove Theorem 1.

*Proof of Theorem 1.* We apply Lemma 11 recursively.

$$\sum_{k=1}^{K}(V_1^k - V_1^{\pi^k})(x_1^k)$$

$$\leq (H+1) + \sum_{k=1}^{K}\left(1+\frac{1}{H}\right)\left(\xi_2^k + \mathrm{clip}\left[\beta_{n_1^k} \mid \frac{\mathrm{gap}_1(x_1^k, a_1^k)}{H+1}\right]\right) + \left(1+\frac{1}{H}\right)^2\sum_{k=1}^{K}(V_2^k - V_2^{\pi^k})(x_2^k)$$

$$\leq \sum_{h=1}^{H} H\left(1+\frac{1}{H}\right)^{2h-1} + \sum_{h=1}^{H}\left(1+\frac{1}{H}\right)^{2h-1}\sum_{k=1}^{K}\left(\xi_{h+1}^k + \mathrm{clip}\left[\beta_{n_h^k} \mid \frac{\mathrm{gap}_h(x_h^k, a_h^k)}{H+1}\right]\right)$$

$$\leq 9H^2 + 9\sum_{h=1}^{H}\sum_{k=1}^{K}(\mathrm{clip}\left[\beta_{n_h^k} \mid \frac{\mathrm{gap}_h(x_h^k, a_h^k)}{H+1}\right] + \xi_{h+1}^k)$$

Note that $\sum_{h=1}^{H}(1+1/H)^{2h-1} \leq \sum_{h=1}^{H}\left((1+1/H)^H\right)^2 \leq e^2 H \leq 9H$. Finally,

$$\sum_{k=1}^{K}(V_1^k - V_1^{\pi^k})(x_1^k) \leq 9H^2 + 9\sum_{h=1}^{H}\sum_{k=1}^{K}(\mathrm{clip}\left[\beta_{n_h^k} \mid \frac{\mathrm{gap}_h(x_h^k, a_h^k)}{H+1}\right] + \xi_{h+1}^k)$$

$$\leq 9H^2 + 18\sqrt{2H^3 K \log(4HK/\delta)} + \sum_{h=1}^{H} 288(\sqrt{H^3 \log(4HK/\delta)} + Ld_{\max})$$

$$\times \inf_{r_0 \in (0, d_{\max}]}\left(\sum_{r=d_{\max}2^{-i}, r \geq r_0} N_r^{\mathrm{pack}}(\mathcal{P}_{h,r}^{Q^\star})\frac{d_{\max}}{r} + \frac{Kr_0}{d_{\max}}\right)$$

$$= \tilde{O}\left(H^{3/2}\inf_{r_0 \in (0, d_{\max}]}\left(\sum_{h=1}^{H}\sum_{r=d_{\max}2^{-i}, r \geq r_0} N_r^{\mathrm{pack}}(\mathcal{P}_{h,r}^{Q^\star})\frac{d_{\max}}{r} + \frac{Kr_0}{d_{\max}}\right)\right)$$

$$+ \tilde{O}\left(H^2 + \sqrt{H^3 K \log(1/\delta)}\right). \qquad \square$$