[Reviews · NeurIPS 2020]

Review 1

Summary and Contributions: The paper focuses on model free RL in continuous state-action spaces endowed with a metric (i.e. distance) function. It proposes a new analysis of an existing (Sinclair et al. 19) optimistic Q-learning algorithm using adaptive partitioning. In particular, under assumptions of smoothness w.r.t. to the metric the authors show that (mostly) without modifications Q-learning with adaptive partitioning achieves a regret that scales with the zooming dimension of the continuous space. This improves over the original analysis of the algorithm that only bounded regret with the covering dimension of the space. The main technical tool used in the new analysis is a regret clipping argument, that allows to obtain an improved recursion and a tighter bound.

Strengths: Zooming dimension can be much smaller than doubling dimension (especially in presence of many sub-optimal actions). In addition the algorithm does not need to be heavily modified to obtain the result. The bound obtained is always tighter than existing bounds, but at the cost of the extra metric smoothness assumption. In simpler (e.g. tabular or bandit) cases the analysis also recover some version of existing bounds, showing both that the analysis is not too loose, as well as a natural generalization of existing work.

Weaknesses: The contribution is sound and well presented, but also does not seem like it will lead to further development or leaves many open questions. While it is positive that the authors show that existing algorithms (i.e. Sinclair et al. 19) already achieved the smaller regret, their new analysis does not seem to be applicable to any other method, or give us any insight on what bottlenecks are left for adaptive partitioning in RL and what could be improved next. Similarly, on a technical level the proof seems to consist mostly of taking existing decomposition and polishing them under the metric smoothness assumption. While the detailed explaination in terms of credit re-assignment is interesting, the approach seems quite incremental and without large further implications. Overall I think the paper is sufficiently polished and interesting to be acceptable, but the lack of potential future development and impact makes it a very borderline submission.

Correctness: Yes

Clarity: Very well written

Relation to Prior Work: Yes, with insight and very fair comparison.

Reproducibility: Yes

Additional Feedback: ===== AFTER REBUTTAL ===== In the rebuttal the authors point out several interesting open questions related to the setting. This increases the impact of disseminating new results on this topic, such as the ones presented in this paper. However they do not really motivate, even at a high level, how their specific contributions open new possibilities for these questions. Overall the rebuttal confirms that the paper is acceptable, but does not increase its strength to the point where I would raise my score.


Review 2

Summary and Contributions: This paper gives a more refined analyses of Adaptive Q-learning algorithm of Sinclair, Banerjee and Yu (2019) by incorporating the notion of the zooming number/dimension, which incorporates more problem-dependent information about the optimal Q/V functions than a covering number over the state-action space. They show that the regret up to time K scales with K^{(z + 1)/(z + 2)} where z is the zooming dimension. Update after rebuttal: No change.

Strengths: Given that the theory is correct, this seems like a significant step towards more nuanced and problem-dependent bounds for MDP with metric spaces.

Weaknesses: There does not seem to be any outstanding weaknesses, given the context that this is a theory paper.

Correctness: To the best of my ability, after a shallow look through the theory in the appendix, the major points all make sense. I'm not familiar enough with the details of the analyses (specifically the metric space portions) to confidently say that it is correct.

Clarity: Other than a lack of prior experience with some of the analyses, everything else was clear. The paper is well written.

Relation to Prior Work: This paper builds a lot on prior analyses, combining ideas from different prior works. The paper is very clear about which points come from where, and how this is different from prior work.

Reproducibility: Yes

Additional Feedback:


Review 3

Summary and Contributions: This paper studies reinforcement learning (RL) problems on large state and action spaces that are endowed with a metric. They key assumption is that the optimal state-action value function, Q*, is Lipschitz smooth with respect to that metric. The setting is that of an episodic, H-stage Markov decision process, in which the learner must choose an action for each stage of each episode while achieving low regret against an optimal policy. Previously, an algorithm was proposed for this problem based on learning the Q* function on an adaptive discretization of the state-action space that becomes steadily finer on important regions of the space. The regret of this algorithm was thought to depend on the packing number of the state-action space. The main contribution of this paper is to improve the analysis of that algorithm, showing that its regret depends on the packing number of a smaller subspace: the set of state-action pairs s, a where Q*(s, a) is close to V*(s). This could be a significant improvement in MDPs where the almost-optimal actions lie in a low-dimensional subspace of the full state-action space. The technique used is adapted from gap-dependent regret bounds for Q-learning in tabular multi-stage MDPs. The packing number of the almost-optimal actions is called the "zooming number" and was previously used to bound regret in contextual Lipschitz bandits. This work therefore shows that the zooming number is a quantity of interest not just in bandits but also in multi-stage MDPs.

Strengths: This paper poses and solves a natural, well-motivated problem. The algorithm considered has been previously proposed and is a straightforward extension of Q-learning to adaptive discretizations of metric spaces. This paper generalizes gap-dependent regret to this non-tabular MDP setting, while also generalizing the zooming dimension from contextual bandits to MDPs.

Weaknesses: As alluded to by the question in the "additional feedback" below, I would have liked to see a simple example of a discrete MDP with a non-trivial metric, i.e. not the tabular MDP. Indeed, casting linear function approximation in this setting would be an excellent aid to intuition, since that setting has been extensively studied.

Correctness: I briefly checked the proofs in the appendix and found no errors. I did not carefully check every line though, so it is possible I might have missed something. There were some small typos: * The variable t is defined in line 7 of the algorithm and 204 of the text, but not subsequently used. On line 10 of the algorithm, t is used instead of n. To reduce the number of symbols defined, I would suggest dropping t and using n everywhere. * "if then" on line 10 of the algorithm is incorrectly formatted.

Clarity: This paper is very straightforward and clearly written. I especially appreciated the informal proof sketch and its discussion of the key arguments.

Relation to Prior Work: The prior work section succeeds in laying out the main threads of research that have contributed to the results in this paper.

Reproducibility: Yes

Additional Feedback: It was instructive to instantiate this result for tabular MDPs and for contextual bandits, and another interesting case is linear function approximation for finite but large state and action spaces. Assume that the optimal Q* function is known to lie in a d-dimensional subspace of the SA-dimensional vector space and the metric comes from the feature representation. Naively, it seems that the algorithm's regret is then O(K^{(d+1)/(d+2)}). Is this "real" or is this type of metric-space algorithm not a good fit for this setting? More generally, is there any evidence that the analysis in this paper is tight for this problem, or even for this algorithm? *** Post-rebuttal Thanks to the authors for their thoughtful response addressing the questions I raised. I agree that it is an important open question whether the linear Q* assumption is enough to get poly(d) algorithms, whether one needs the stronger linear MDP assumption, and furthermore if metric space algorithms can also achieve this under further assumptions.


Review 4

Summary and Contributions: This paper studies the episodic MDP in metric space and shows UCB-Q learning in continuous RL space is provably efficient with the zooming dimension.

Strengths: The theoretical results are soundness and the results are improved compared to prior work.

Weaknesses: The main difference of results between this paper and Sinclair et al. (2019) is the replacement of covering dimension to zooming dimension. But there is only one trivial example to illustrate the difference, which is not clear and sound enough. I recommend the authors pay more attention to this part, which is the improvement compared to prior works. For example, the authors could construct a few examples of continuous MDP to show the difference between zooming dimension and covering dimension, or give some soundly theoretical results instead of intuition. Besides, I'm confused by the clipped trick. As the paper claimed the clipped trick can solve the impossibility of bounding the number of active balls. Could you explain it in mathematical form?

Correctness: 1. Algorithm 1, Line 11, missing '1/2 r(B)-net' 2. Theorem 2, missing plus '+'

Clarity: This paper is well written but the content of Sec 5(proof sketch) is not well-organized nor clear enough.

Relation to Prior Work: yes

Reproducibility: Yes

Additional Feedback: After reading authors' respones, I decide to raise my score to 6, as the authors' feedback solves my confusions.

[Author Response · NeurIPS 2020]

We would like to thank all the reviewers for the detailed and insightful comments.

**Reviewer 1**

*Regarding applying our technique to other methods*: The clipping trick and much of the subsequent analysis can be
used to analyze other methods, including UCB-VI, UBEV, and other model-based methods. In fact, Simchowitz and
Jamieson (2019) use the clipping trick to analyze model-based algorithms in the tabular setting, not the model-free one
we analyze here. We do not include this in our paper because the model-based algorithms incur an $S^2$ dependence in
the "lower order" $K$-independent term. This degrades the rate substantially in the nonparametric case, since effectively
we set $S$ in terms of the number of episodes $K$ and the zooming dimension. So this term is not actually "lower order"
in the nonparametric case, which leads to a worse final bound.

*Regarding future work*: Based on the above remarks, an intriguing question is whether model-based methods can
achieve similar "zooming" guarantees via adaptive partitioning. A very recent paper of Sinclair et al. (`https:`
`//arxiv.org/abs/2007.00717`) make some progress in this direction, but does not obtain zooming-dimension
dependence. We also mention two other questions: Can our techniques and adaptive guarantees extend to the infinite
horizon discounted setting? Can adaptive discretization be combined with function approximation, analogous to the
policy zooming approach of Krishnamurthy et al. (2019) for contextual bandits? This would allow us to use function
approximation to deal with large state spaces and adaptive discretization to deal with large action spaces, which may be
quite effective in practice. We can mention these directions in the final version.

**Reviewer 2.** Thanks for the kind words!

**Reviewer 3.** Thanks for catching the typos. We will fix them in the final version.

*Regarding linear methods*: This is a great question, but quite subtle! Remember, our method applies when $Q^\star$ is
*Lipschitz* continuous. This is of course implied by $Q^\star$ being linear in some known features (assuming boundedness), but
linearity is a much stronger assumption. A helpful analogy to think about is regression: if one knew the true regression
function were linear, one should not use a nonparametric method. But nonparametric methods are applicable much
more generally. It's the standard estimation error/approximation error tradeoff.

However, the RL setting is more subtle. If $Q^\star$ is Lipschitz, then work from Lipschitz bandits suggests that our method
is optimal. If $Q^\star$ is linear (and that is all we assume), then our method acheives the $K^{\frac{d+1}{d+2}}$ regret rate, but actually we
do not know of any better guarantees for this setting. In particular, the recent $\mathrm{poly}(d)\sqrt{K}$ results for linear function
approximation require much stronger assumptions, such as linear MDP or low inherent Bellman error. $Q^\star$ linear is a
much weaker assumption than e.g., linear MDP, and it is not clear whether $\mathrm{poly}(d)\sqrt{K}$ rates are achievable here at all.

**Reviewer 4** Thanks for catching the typos. We will fix them in the final version.

*Regarding zooming dimension vs covering dimension*: We encourage the reviewer to examine the experiments of
Sinclair et al., (2019), which shows (1) that adaptive partitioning performs much better than uniform discretization and
(2) provides concrete examples and visualizations of this benefit. The zooming dimension gives a tighter bound for
Sinclair et al.,'s adaptive partitioning scheme and captures the improvements observed in the experiments. Note that we
analyze exactly their algorithm but their bound does not capture these improvements.

In addition, (1) the example we give in Figure 1 of our paper is actually quite general, as any problem where the
near-optimal actions concentrate onto a low dimensional manifold has smaller zooming dimension than covering
dimension; (2) Theorem 1 is even more refined than the zooming dimension bound; (3) zooming dimension and
related quantities like "near optimality dimension" are widely studied in the bandit literature (20+ papers in the top ML
conferences) and it is natural to extend these notions to RL.

*Regarding the clipping trick*: Please see Lemma 11 and 13 in the supplement for a mathematical statement. In words,
instead of trying to bound the number of balls (which would be quite large), we notice that the regret incurred is
determined by the sum of *clipped* surpluses for this ball, where we clip at level $\mathrm{gap}/(H+1)$. This is because we only
play a bad ball when there is a large error at a later time step, so we can credit this mistake to the later error and clip the
current surplus. Using this decomposition, clipping will quickly take effect for "bad" balls, which allows us to bound
the regret incurred by this ball.

[Meta-Review · NeurIPS 2020]

This paper is about model-free RL where the state-action state is a metric space. An improved analysis of an existing algorithm (with some modifications) is shown to achieve a regret that scales with the zooming dimension of the metric space, instead of the covering dimesion. A general consensus among reviewers emerged that this theoretical RL paper is well executed, and provides a reasonable though not groundbreaking contribution to the RL literature.